

# Agronomical, biochemical and histological response of resistant and susceptible wheat and barley under BYDV stress

Shormin Choudhury[1,2], Hongliang Hu[1], Philip Larkin[3], Holger Meinke[1,4], Sergey Shabala[4], Ibrahim Ahmed[1] and Meixue Zhou[1]

[1] Tasmanian Institute of Agriculture, University of Tasmania, Prospect, Tasmania, Australia
[2] Department of Horticulture, Faculty of Agriculture, Sher-e-Bangla Agricultural University, Sher-e-Bangla Nagar, Dhaka, Bangladesh
[3] CSIRO Agriculture and Food, Canberra, Australia
[4] Tasmanian Institute of Agriculture, University of Tasmania, Hobart, Tasmania, Australia

## ABSTRACT

Barley yellow dwarf virus-PAV (BYDV-PAV) is one of the major viruses causing a widespread and serious viral disease affecting cereal crops. To gain a better understanding of plant defence mechanisms of BYDV resistance genes (*Bdv2* and *RYd2*) against BYDV-PAV infection, the differences in agronomical, biochemical and histological changes between susceptible and resistant wheat and barley cultivars were investigated. We found that root growth and total dry matter of susceptible cultivars showed greater reduction than that of resistant ones after infection. BYDV infected leaves in susceptible wheat and barley cultivars showed a significant reduction in photosynthetic pigments, an increase in the concentration of reducing sugar. The protein levels were also low in infected leaves. There was a significant increase in total phenol contents in resistant cultivars, which might reflect a protective mechanism of plants against virus infection. In phloem tissue, sieve elements (SE) and companion cells (CC) were severely damaged in susceptible cultivars after infection. It is suggested that restriction of viral movement in the phloem tissue and increased production of phenolic compounds may play a role in the resistance and defensive mechanisms of both *Bdv2* and *RYd2* against virus infection.

## INTRODUCTION

Barley yellow dwarf virus-PAV (BYDV-PAV) is transmitted by aphids and has been recognized as one of the most serious viral pathogens of the *Luteovirus* genus that systemically infects cereal crops (*Jiménez-Martínez et al., 2004*). Infection with BYDV-PAV causes significant economic losses throughout the world (*Huth, 2000*; *Ramsell et al., 2008*). The use of resistant or tolerant varieties is an effective solution and economical method for controlling BYD disease (*Ordon et al., 2004*). Virus tolerance is the capability of a host genotype to survive or recover from the damaging effects of virus infection and yield, while resistance is the plant's ability to restrict or prevent the infection of virus (*Cooper & Jones, 1983*). Wheat lines containing *Bdv2* gene showed less yellowing and lower viral

Corresponding author
Meixue Zhou, mzhou@utas.edu.au, meixue.zhou @utas.edu.au

titer than susceptible wheat lines when infected by BYDV (*Kausar et al., 2015*). Similarly, barley cultivars containing *Ryd2* also have lower virus titre after BYDV infection, which leads to less visual symptom and grain yield reduction (*Beoni et al., 2016*). Visual scoring of symptoms for BYDV-PAV resistance is not always useful as BYDV-PAV can multiply and spread in the plant without showing any visual symptoms (*Horn, Habekuß & Stich, 2013*). Whole-plant metabolite profiles can be altered by virus infection (*Shalitin & Wolf, 2000*; *Xu et al., 2008*). Through virus infection, many plant defence pathways can be activated or suppressed (*Lewsey et al., 2010*; *Whitham, Yang & Goodin, 2006*).

BYDV is transmitted in a persistent, circulative and non-propagative manner (*Conti et al., 1990*; *Masterman, Holmes & Foster, 1994*) and its transmission occur when an aphid feeds on infected phloem and phloem cells and then transfers the viruses in its saliva to healthy plants (*Walling, 2008*). Virus spread usually starts from cell-to-cell (short distance movement). In the later phase, the virus enters into the vascular tissue, where it is transported rapidly via phloem cells. This is referred to as long distance movement (*Hipper et al., 2013*; *Waigmann et al., 2004*). Plants infected by virus undergo strong metabolic and ultrastructural changes, even when no visible symptoms are apparent (*Yan et al., 2008*). Disease development in host plants is likely to induce substantial biochemical changes such as in protein, phenolics, carbohydrates, and these metabolic changes may favour or inhibit disease development (*Ayres, Press & Spencer-Phillips, 1996*). In certain plant host-pathogen interactions, these alterations may play a major role in contributing to disease resistance. In crop breeding, the response of biochemical compounds in plants has been helpful to select fungal and insect resistant genotypes (*Lattanzio, Lattanzio & Cardinali, 2006*).

In chilli plants inoculated with pepper leaf curl virus (PepLCV), total phenolic content was increased in resistant cultivars and was decreased in susceptible cultivars (*Rai et al., 2010*). A positive correlation was observed in cocoa between cocoa swollen shoot virus disease (CSSVD) resistance and total phenolic contents at 3 months after inoculation (*Ofori et al., 2015*). Rapid synthesis of phenolics and their polymerization in the cell wall has been suggested as a plant defence response against infection (*Sattler & Funnell-Harris, 2013*; *Matern & Kneusel, 1988*), while low levels of phenolics may be linked to disease susceptibility (*Yao, De Luca & Brisson, 1995*). However, there is no report on the relationship between total phenolic contents measured after BYDV infection and BYDV resistance in cereal crops.

Sugar metabolism is a dynamic process with both metabolic fluxes and sugar concentrations fluctuating strongly throughout plant development and in response to environmental signals for example circadian changes and biotic stresses (*Bläsing et al., 2005*; *Borisjuk et al., 2003*). In melon plants, cucumber mosaic virus infection causes a significant increase in the sugar content within the phloem (*Shalitin & Wolf, 2000*). An increase in sugar concentration in tobacco leaves was caused by potato leafroll virus (PLRV) infection inhibiting phloem loading; the increased sugar led to the inhibition of photosynthesis (*Herbers et al., 1997*). Reduced translocation of sugar and other nutrient molecules to the root system limits root growth and function and thus affects plant growth and grain yield (*Riedell et al., 2003*). Biotic stress can also inhibit chlorophyll synthesis (*Funayama-Noguchi & Terashima, 2006*; *Šutić & Sinclair, 1991*), resulting in reduced photosynthesis. To date,

many studies have been done with BYDV but little information has been reported regarding changes of biochemical compounds caused by BYDV infection in wheat and barley.

The aim of the study was to assess the response of different agronomical, biochemical and cell ultrastructural changes after systemic BYDV infection of susceptible and resistant wheat and barley plants, and to provide a better understanding of resistance mechanisms against BYDV-PAV infections.

## MATERIALS AND METHODS

### Plant material

Two wheat (*Triticum aestivum*) cultivars (Manning and Revenue) and two barley (*Hordeum vulgare*) cultivars (Franklin and Flagship) were used in this study. Manning and Franklin are the cultivars with known BYDV resistance, containing *Bdv2* (*Ayala-Navarrete et al., 2007*) and *Ryd2* gene (*Raman & Read, 1999*), respectively.

### Aphid colonies

A colony of bird-cherry aphid, *Rhopalosiphum padi*, was collected from a Tasmanian barley field trial in 2014, and reared on barley (cv. TAM407227-a BYDV susceptible genotype) in a cage at 20 °C $\pm$ 2 °C, 65 $\pm$ 5% RH, with a photoperiod of L14:D10 using cool white fluorescent light under 450 $\mu$mol-m$^{-2}$ s$^{-1}$ photosynthethically active radiation (PAR).

### Virus isolates

One isolate of BYDV-PAV was obtained from the University of New England, New South Wales (NSW), Australia and maintained in barley cv. TAM407227 in small cages under the same conditions as the aphid colonies. The virus isolate was periodically (6-weekly) transferred to new plants using *R. padi* in clip cages.

### Plant growth and virus inoculation

Ten seeds of each cultivar were sown in 2 l plastic pots, which filled with pre-fertilized potting mixture. After germination, seedlings were thinned to five uniform and healthy plants in each pot. The plants were grown in a glasshouse, between September and November 2016. The average temperature was 23 °C in daytime and 15 °C at night with a relative humidity of 65 to 80%. At two-leaf stage, each plant was inoculated with BYDV-PAV using ten viruliferous adult aphids (*Rhopalosiphum padi*) in a clip cage. An inoculation access period of 120 h was used to ensure virus infection of all plants. Aphids were then killed by spraying 1 ml/L solution of the insecticide Karate (Syngenta Ltd.).

### Leaf samples for biochemical analyses

The most recent fully expanded leaves of both controls and inoculated plants were harvested at different growth stages, i.e., three and six weeks after inoculation (WAI), for various analysis. All biochemical parameters were measured using spectrophotometer (Genesys 10S UV-Vis).

#### Photosynthetic pigments

Photosynthetic pigments were measured using the method of (*Moran & Porath, 1980*). 0.2 g leaf tissue was ground into powder with liquid nitrogen, then homogenised with 1 ml

100% N, N-dimethylformamide (DMF). Homogenized samples were centrifuged at 10,000 x g for 10 min to gather the supernatant. Then 1 ml DMF was added again and samples were centrifuged. The supernatant was removed and another 1 ml DMF was added. The absorbance was recorded at 663 and 645 nm in a spectrophotometer. Calibration was done by using a blank of 100% DMF. Chlorophyll a, b and total chlorophyll were calculated by following formulas:

$$\text{Chlorophyll a (mg g}^{-1}\text{tissue)} = \frac{[12.7\,(OD663) - 2.69\,(OD645)]/\times V}{1000} \times W$$

$$\text{Chlorophyll b (mg g}^{-1}\text{tissue)} = \frac{[22.9\,(OD645) - 4.68\,(OD663)]/\times V}{1000} \times W$$

$$\text{Total Chlorophyll (mg g}^{-1}\text{tissue)} = \frac{[8.02\,(OD663) + 20.20\,(OD645)]/\times V}{1000} \times W.$$

Where OD, Optical density at respective nm, V, Final volume of chlorophyll extract, W, Fresh weight of the tissue extracted.

### Measurement of total protein content

Total protein was estimated by using Bradford method (*Bradford, 1976*) and absorbance was recorded at 595 nm. Bovine serum albumin was used as standard. Protein contents in leaf samples were recorded as $\mu$g of protein per gram of leaf tissue.

### Phenolic content

Phenol content was measured using the method of *Singleton, Orthofer & Lamuela-Raventós (1999)*. Fresh leaves (250 mg) were homogenized with 85% methanol. The extract was centrifuged at 3,000 g for 15 min at 10 °C and the supernatant was separated. Folin & Ciocalteu's reagent (2 ml) was added to each 2 ml of the supernatant. A sodium carbonate solution (7.5%, 2 ml) was added to each test tube and after 30–45 min, the absorbance was read at wavelength 725 nm against a reagent blank. A standard curve using gallic acid was generated to determine the concentration of total phenols in the unknown sample.

### Reducing sugars content

Reducing sugars were determined based on the method of phenol-sulphuric acid (*DuBois et al., 1956*). A total of 0.2 g fresh leaf was homogenized with deionized water and the extract was filtered. 2 ml of the solution were mixed with 0.4 ml of 5% phenol. Subsequently, 2 ml of 98% sulphuric acid were added rapidly to the mixture. The test tubes were allowed to keep for 10 min at room temperature, and placed in a water bath at 30 °C for 20 min for colour development. Light absorption at 540 nm was then recorded with the spectrophotometer. Blank solution (distilled water) was prepared in the same way as above (*Ammar et al., 2009*). Contents of reducing sugar was expressed as mg g$^{-1}$ fresh weight (FW).

## Biomass production

Four plants (above ground) were randomly sampled from each treatment and replication at 6 WAI. After taking the fresh weight plant samples were kept in oven at 65 °C for 72 h before recording the weight of dry matter.

## Enzyme-linked immunosorbent assay (ELISA)

Leaves from four plants of each treatment and replication were collected at 6 WAI for ELISA test. BYDV-PAV polyclonal antibodies (Sediag, Colmar, France) were used in DAS-ELISA (*Clark & Adams, 1977*) to detect the virus in leaf tissues. Samples were prepared by grinding 1 g leaf tissue in phosphate buffered saline, pH 7.4, with 2% polyvinylpyrrolidone and 0.2% egg albumin in a ratio of 1:20. We used 2 healthy controls and 2 positive controls. All samples (control and BYDV-PAV inoculated leaves) and positive and negative controls were tested in duplicate. Microplates were read using a photometer (MR 5000; Dynatech; Melville, NY, USA) at wavelength 405 nm. Our ELISA cut-off value is 2 times of the negative control (healthy control) in each test. Samples with absorbance values greater than twice the mean of negative controls were considered positive (*Clark & Adams, 1977*).

## Histological examination

Anatomical structure of infected and control wheat and barley leaves was examined with a light microscope using Leica DM500 (Buffalo Grove, IL, USA). Three biological replications were performed for each treatment. For microscopic examination, wheat leaves ($2 \times 2$ mm$^2$) from both susceptible and resistant leaves were cassetted (Techno Plas, St. Mary's, South Australia) using biopsy pads (Trajan Scientific and Medical, Victoria, Australia). Samples were then fixed in 10% neutral buffered formaldehyde (Confix, ACFC, Australian Biostain; Traralgon, Victoria, Australia) for 24 h and processed overnight using a standard 15 h overnight procedure in an ASP300S tissue processor (Leica Microsystems, Wetzlar, Germany). Samples were orientated on the EG1160 (Leica), embedded in paraffin wax (Surgipath Paraplast, 39601006; Leica) and sectioned at 3 microns using Leica RM2245 microtome and adhered to microscope slides (Menzel Glaser, Braunschweig, Germany) for 20 min at 60 °C. Sections were deparaffinised, rehydrated and stained using Jung autostainer XL (Leica) for haematoxylin (Harris' Haematoxylin, AHHNA, Australian Biostain) and eosin, dehydrated cleared and cover-slipped (Leica CV5030) using CV Mount (Leica 046430011).

## Determination of root growth

For root length measurement, five seedlings were grown in a 2 l plastic pot filled with pre-fertilized potting mixture. The plants were grown in a glasshouse with the average temperature of 15 °C in daytime and 8 °C at night. Plant was inoculated with 10 viruliferous aphids for 120 h. The experiment was terminated at 3 WAI and root length was measured. The roots were carefully washed with tap water to separate substrates. The longest root length (cm plant$^{-1}$) was measured as the distance from the base of the plant to the end of the longest root. Five biological replications were performed for each treatment.

## Data analysis

The experiments used a randomized complete block design (RCBD) with three replications for each cultivar and five plants in each replicate. Data were analysed using software SPSS 20.0. Two treatment means (the values of virus infected and control plants) were subjected to paired $t$-test. The value was considered to be statistically significant when $P < 0.05$. All results were presented with mean $\pm$ SE from the replicates. Graphs were drawn using the

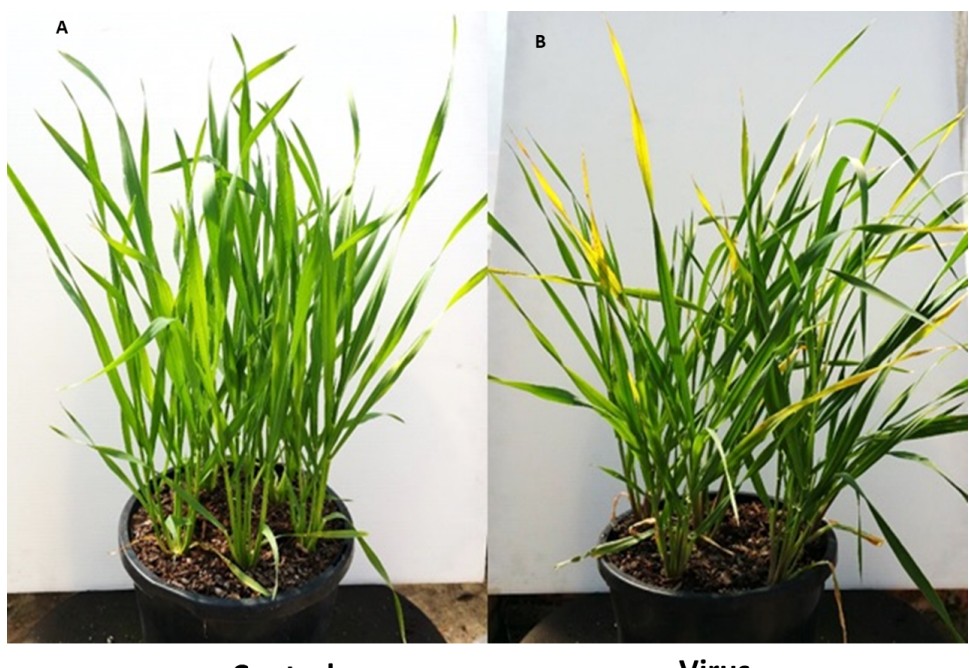

**Control**    **Virus**

**Figure 1    BYDV-PAV inoculated and control plants of susceptible barley cultivar Flagship at 3 WAI.**
(A) Control. (B) Virus infected.

Microsoft Excel program. We used ANOVA to test the effect of BYDV stress, cultivars and their interaction on biochemical parameters.

## RESULTS

### Symptoms after inoculation

Typical symptoms appeared on virus infected plants included leaf discoloration and dwarfism. Leaf discoloration in both inoculated barley cultivars was visible within 3 WAI (Fig. 1). At 6 WAI we did not observe leaf discoloration in either of the wheat cultivars, although the susceptible cultivar showed evidence of dwarfism at that time (Fig. 2).

### Validation of inoculation

ELISA was used to confirm the virus infection of a plant, when ELISA values above the detection threshold (A 405 > 0.32) were assumed to illustrate the presence of virus particles. The virus extinction value was the highest in BYDV inoculated susceptible barley plant (Flagship) followed by susceptible wheat (Revenue) and the lowest value was detected in resistant wheat variety (Manning) (Fig. 3), suggesting that both the *Ryd2* gene of barley and the *Bdv2* gene of wheat reduced the viral load.

### The effect of virus infection on root growth

At 3 WAI, all inoculated wheat and barley cultivars showed reduced root length compared to the control (Figs. 4A–4D). Susceptible barley and wheat cultivars showed significantly greater reduction of root length by 41% and 36% for Revenue and Flagship, respectively.

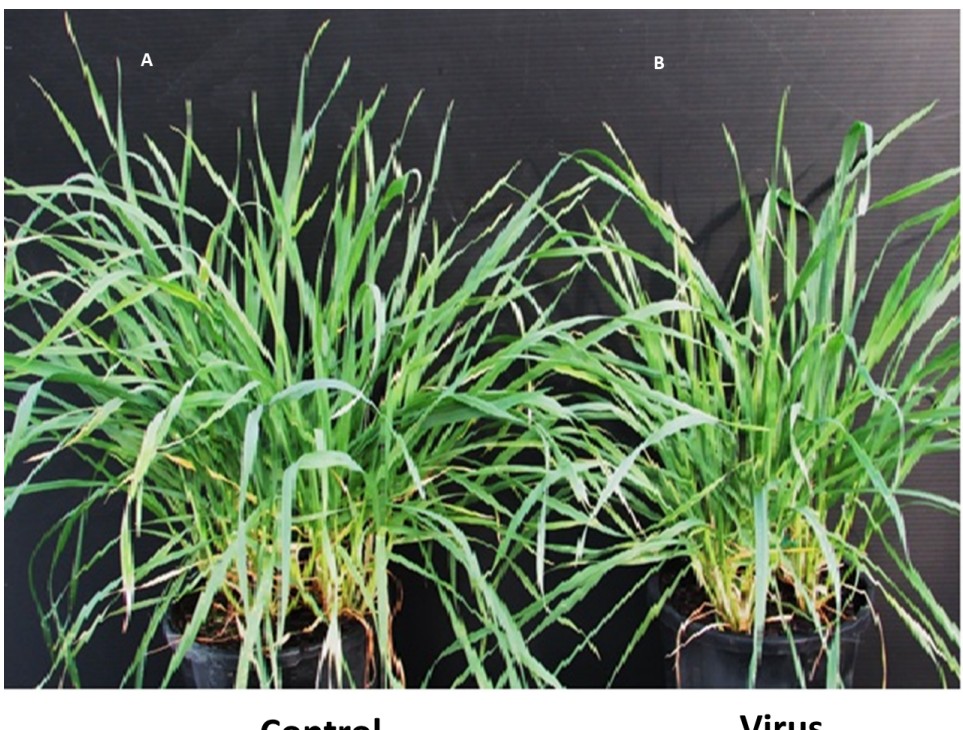

**Figure 2** **BYDV-PAV inoculated and control plants of susceptible wheat cultivar Revenue 6 WAI.** (A) Control. (B) Virus infected.

In contrast, resistant barley (Franklin) and wheat (Manning) only exhibited 7% ($p > 0.05$) and 13% ($p < 0.05$) reduction, respectively (Fig. 4E).

## Photosynthetic pigments

At 3 WAI, the contents of photosynthetic pigments were significantly reduced in virus infected plants of both resistant and susceptible cultivars compared to the control. No significant interactions were found in cultivar × treatment for all photosynthetic pigments at 3 WAI (Table S1) even though greater reductions were found in susceptible ones. The average reductions in chlorophyll a, chlorophyll b and total chlorophyll were 33%, 50%, and 38%, respectively for Revenue, and 24%, 38% and 28%, respectively, for Flagship (Figs. 5A–5C).

At 6 WAI, the difference in photosynthetic pigments between inoculated and control plants of the two resistant wheat and barley cultivars were insignificant. In contrast, further reductions in photosynthetic pigments were found in the two susceptible cultivars (Figs. 5D–5F and Table S1).

## Total protein

In wheat, virus infection caused a significant reduction in foliar protein contents only of susceptible wheat cultivar (Revenue) but not of the other cultivars at 3WAI ($P < 0.05$) (Fig. 6A). At 6 WAI, both susceptible and resistant barley and wheat cultivars showed
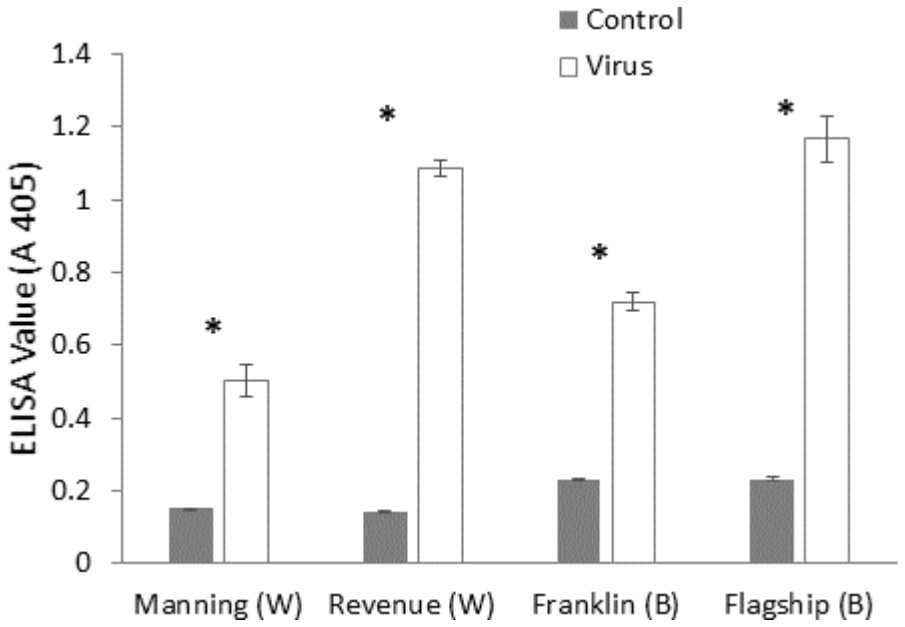

**Figure 3 Mean virus extinction (A 405 nm) assessed in leaf extracts of BYDV-PAV inoculated and control plants of wheat (W) and barley (B) cultivars at 6 WAI.** Mean ± SE ($n = 9$).

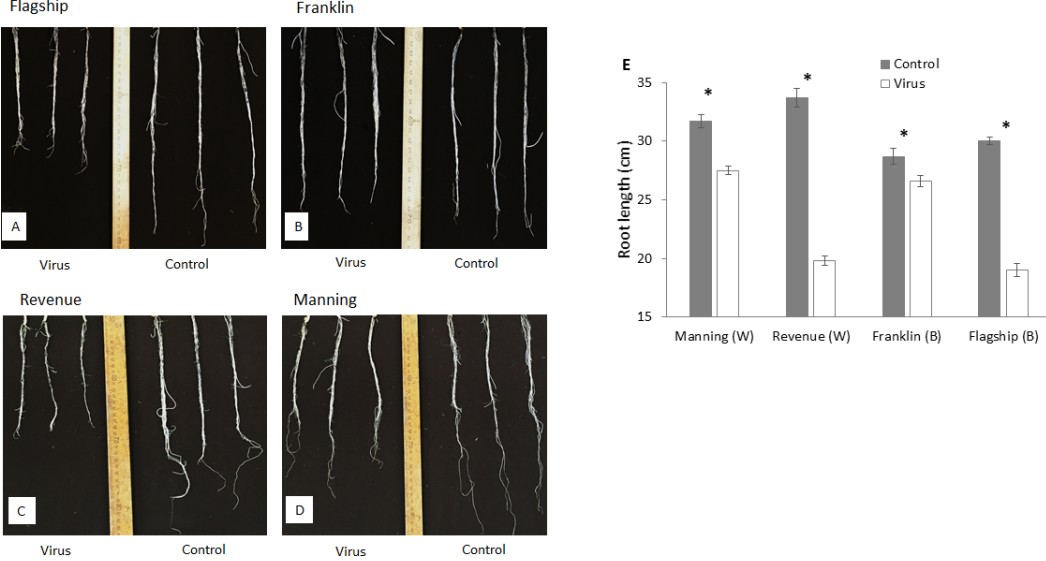

**Figure 4 Root appearance (A–D) and average root length (E) of BYDV-PAV inoculated and control plants of wheat (W) and barley (B) cultivars at 3 WAI.** Mean ± SE ($n = 5$).

significant reduction in protein content in virus infected plants. Significant cultivar × treatment interaction was found on protein content at 6 WAI (Table S1).The reduction was observed more in susceptible cultivars Flagship (28%) and Revenue (27%) whereas the reductions in Manning and Franklin were only 9% and 11%, respectively (Fig. 6B).

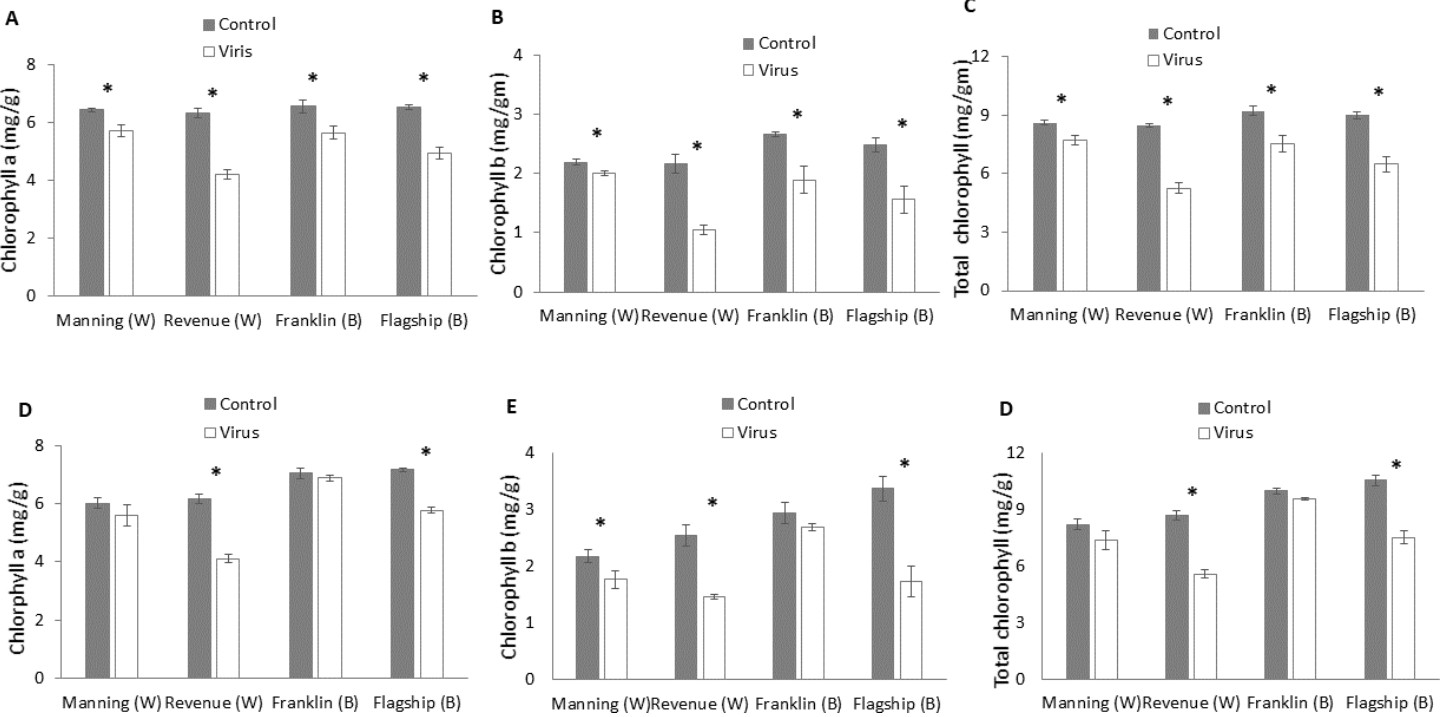

**Figure 5** Average content (mg/g) of chlorophyll a, chlorophyll b, and total chlorophyll of BYDV-PAV inoculated and control plants of wheat (W) and barley (B) cultivars at 3 WAI (A–C) and 6 WAI (D–F). Mean ± SE ($n = 6$).

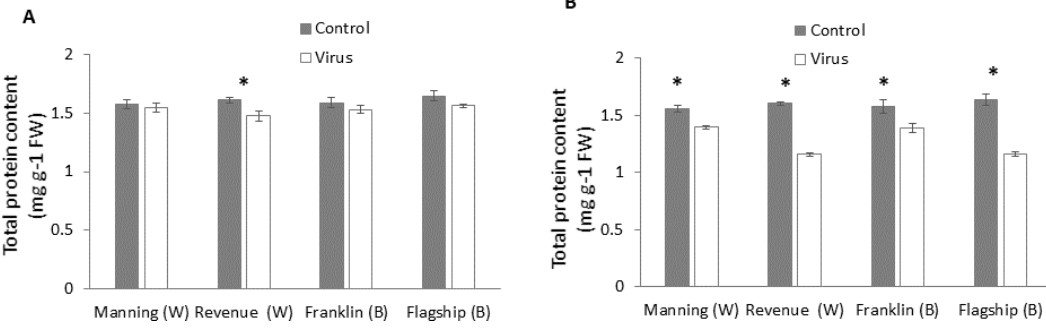

**Figure 6** Average content (mg/g leaf fresh weight) of total protein of BYDV-PAV inoculated and control plants of wheat (W) and barley (B) cultivars at 3 WAI (A) and 6 WAI (B). Mean ± SE ($n = 6$).

## Total phenol

At early stage (3 WAI) of virus infection, significant differences in total phenol contents were found among cultivars (Table S1). However, total phenol contents were not significantly different between virus infected and control plants of all cultivars (Fig. 7A and Table S1).

However, at 6 WAI, significant increases in total phenol contents were found in virus infected plants of all cultivars. The increase in total phenol contents was more pronounced
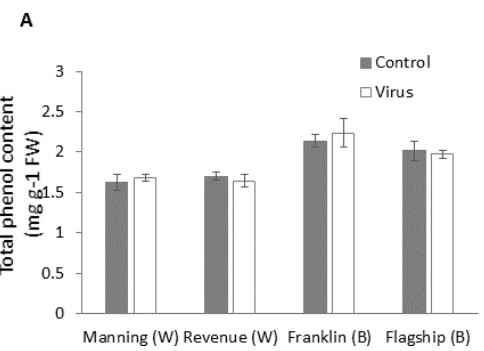
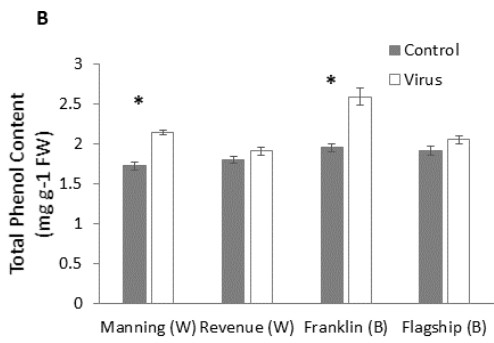

**Figure 7 Average content (mg/g leaf fresh weight) of total phenol of BYDV-PAV inoculated and control plants of wheat (W) and barley (B) cultivars at 3 WAI (A) and 6 WAI (B).** Mean ± SE ($n = 6$).

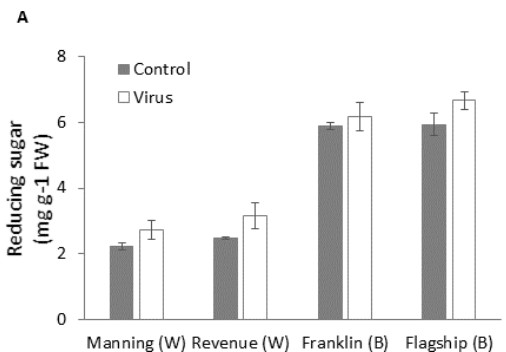
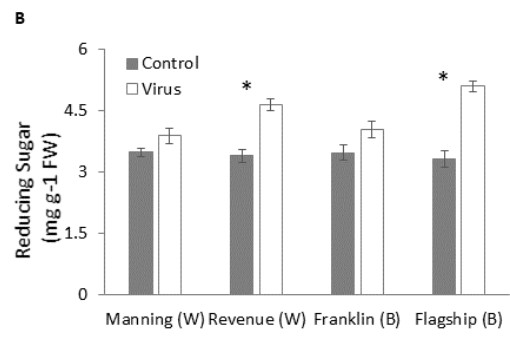

**Figure 8 Average content of (mg/g leaf fresh weight) reducing sugar of BYDV-PAV inoculated and control plants of wheat (W) and barley (B) cultivars at 3 WAI (A) and 6 WAI (B).** Mean ± SE ($n = 6$).

in the resistant cultivars (19–25%) than susceptible ones (5–6%) (Fig. 7B), indicating a significant cultivar × treatment interaction (Table S1).

## Reducing sugar

Figures 8A–8B show that virus infection caused an increase in leaf sugar content of both susceptible and resistant cultivars. Wheat cultivars had lower sugar contents than barley cultivars earlier at 3 WAI but they were similar to the barley cultivars at later stage (6 WAI). Significant cultivar × treatment interactions were found on sugar content at 6 WAI (Table S1). Susceptible cultivars showed a greater increase in reducing sugar contents in virus infected plants than resistant ones, being 9%, 26%, 13% and 35% at 6 WAI for Manning, Revenue, Franklin and Flagship, respectively (Fig. 8B).

## Biomass production

Significant differences in biomass were found in BYDV inoculated and non-inoculated control plants for all cultivars. The reduction of fresh weight varied with cultivar, with the lowest reduction occurring in the resistant wheat cultivar Manning (4%) and the highest in the susceptible wheat cultivar Revenue (41%) (Fig. 9A). A similar trend was found for

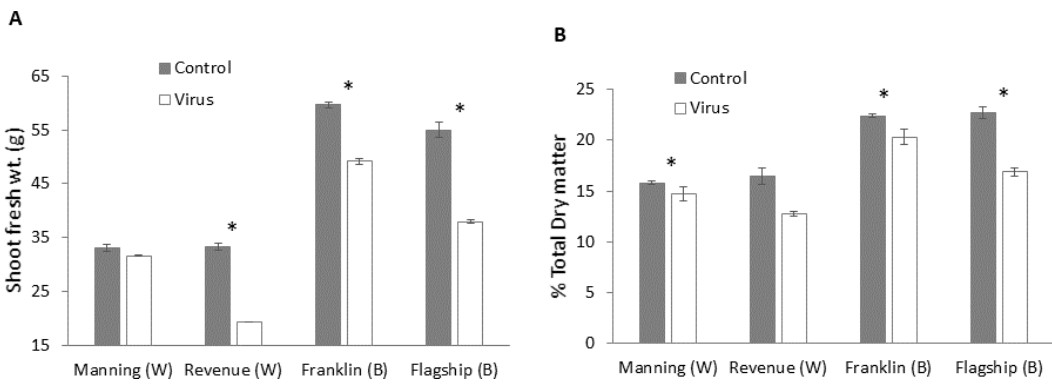

**Figure 9** **Average shoot fresh weight (g) (A) and relative dry matter (B) of BYDV-PAV inoculated and control plants of wheat (W) and barley (B) cultivars at 6 WAI.** Mean $\pm$ SE ($n = 9$).

the dry matter production. Greater reductions in dry matter were observed in Flagship (25%) and Revenue (22%) whereas the reductions in the resistant cultivars Manning and Franklin were only 6% and 9%, respectively (Fig. 9B).

## Alteration of leaf ultrastructure

Ultrastructural examinations of the phloem tissue of leaves from all cultivars were conducted at 6 WAI. The sections of leaf venial regions of non-inoculated plants showed a typical structure of vascular bundles in both wheat and barley. The sieve elements (SE) and the companion cell (CC) are well organised in the phloem tissue with each SE being adjoined by CC (Figs. 10A, 10C and 11A, 11C).

In virus infected barley plants, the phloem tissue of leaf veins consisted of smaller, denser and disorganised SE, with no adjacent CC. In addition, in the susceptible barley cultivar Flagship the SE became necrotic, covered with dark stain and the CCs were degenerated (Fig. 10B). Different results were observed in the resistant cultivar Franklin, which had normal SE with adjacent CC. Although these were looking almost the same as non-inoculated leaves, the CC seems to have reduced size compared with control. Necrotic regions were also observed in some vascular bundles in Flagship (Fig. 10B). In the susceptible wheat, BYDV-PAV inoculated leaf showed infected phloem parenchyma (IPP) and infected sieve elements (ISE) (Fig. 11B). The resistant wheat plant had similar cellular structures of SE and CC in virus infected and control leaves, thus vascular bundle was not affected in virus infected resistant wheat cv. Manning (Fig. 11D).

## DISCUSSION

BYDV-PAV is one of the most destructive diseases of wheat and barley, which often causing significant yield losses when susceptible cultivars are grown (*Jarošová et al., 2013*). To reduce BYD disease damage, the use of resistant cultivars is the most cost-effective and environmentally sound approach. For a better understanding of the mechanism of plant resistance to BYD disease, we investigated the changes of biochemical and ultrastructural characteristics in susceptible and resistant wheat and barley cultivars.

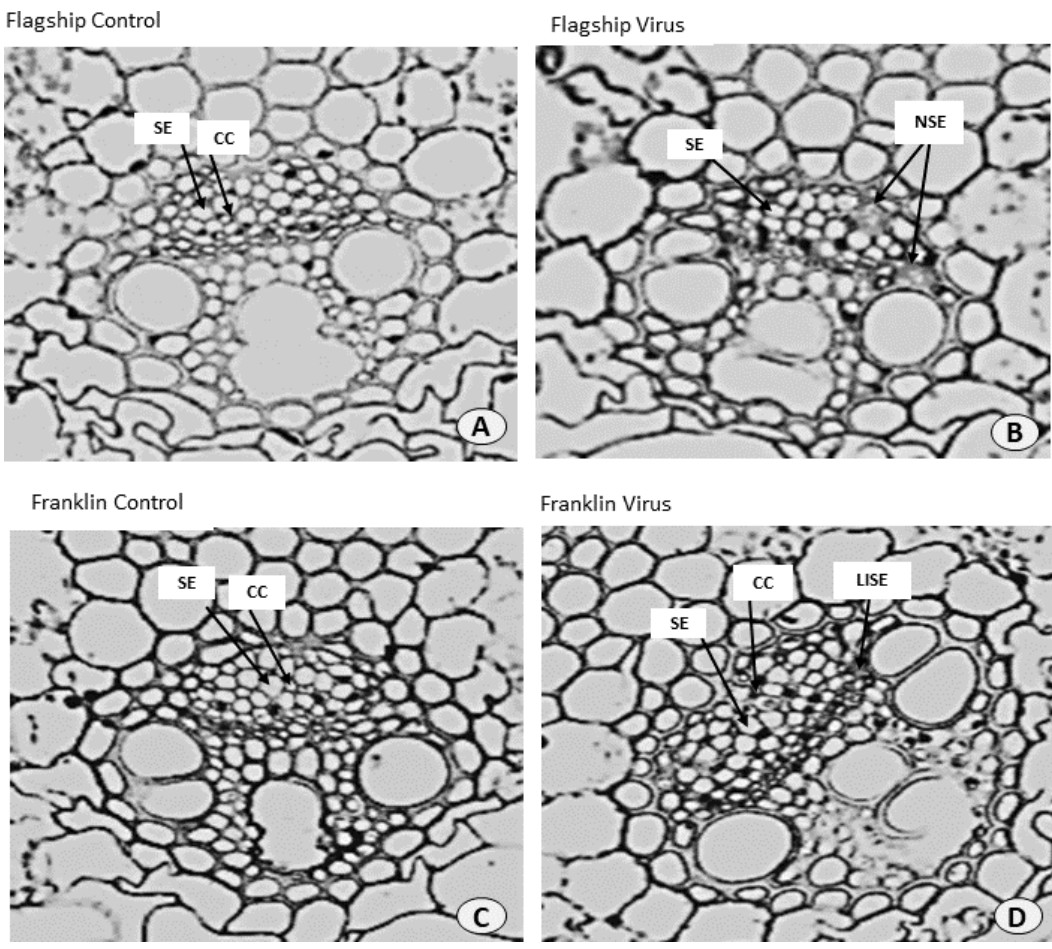

**Figure 10** **Transverse sections of foliar vascular bundles of susceptible (Flagship) and resistant (Franklin) barley cultivars.** (A, C) from control plants and (B, D) from BYDV-PAV inoculated plants. Details are SE, sieve elements; CC, companion cell; NSE, necrotic sieve elements, LISE, little infected sieve elements. Mean ± SE ($n = 3$).

Reduction in chlorophyll content has been reported in many host plants infected with different viruses. The virus infection reduces chlorophyll contents of leaves producing chlorosis (*Pineda et al., 2008*; *Vimla & Shukla, 2009*). In our experiment, at 6 WAI susceptible cultivars showed a significant decrease in the rate of photosynthetic pigments. The reduced chlorophyll contents in susceptible cultivars are mainly due to a loss of leaf photosynthetic area and chloroplast disorders as observed in bean mosaic virus infected *vicia faba* leaves (*Radwan et al., 2008*) while a resistant gene can prevent the loss of chlorophyll in virus-infected leaves as shown in resistant tomato genotype after TMV infection (*Fraser & Loughlin, 1980*). Yellow vein mosaic virus (YVMV) infection can cause enhanced activity of the chlorophyllase that attack chlorophyll and inhibit chloroplast development and chlorophyll synthesis in okra leaves (*Ahmed, Thakur & Bajaj, 1986*). The reduced photosynthesis capacity caused by reduced content of photosynthetic pigments contributes to the decrease in biomass production, which is shown in our studies as well

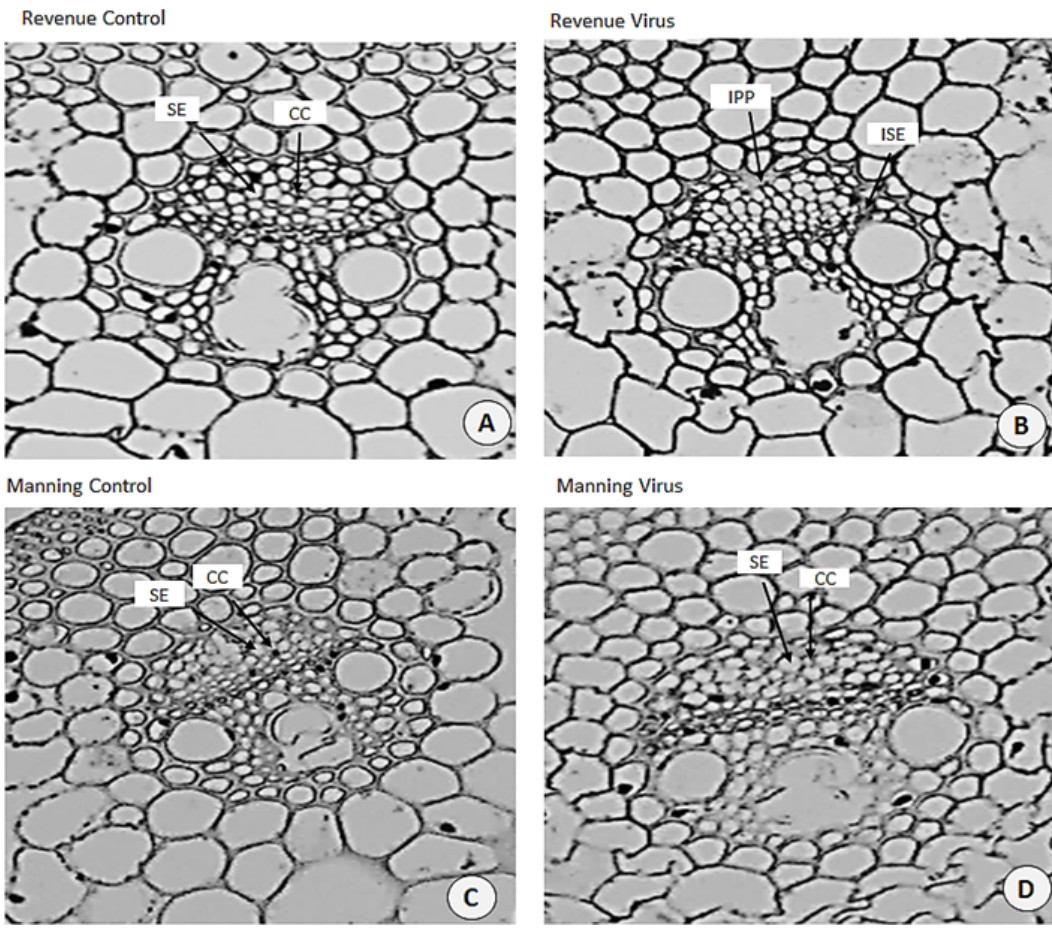

**Figure 11  Transverse sections of foliar vascular bundles of susceptible (Revenue) and resistant (Manning) wheat cultivars.** (A, C) from control plants and (B, D) from BYDV-PAV inoculated plants. Details are SE, sieve elements; CC, companion cell; ISE, infected sieve elements; IPP, infected phloem parenchyma. Mean ± SE (*n* = 3).

as previous reports on BYDV infected cereals (*Bukvayová et al., 2006*). Resistant cultivars infected with BYDV showed little effect on plant biomass, indicated resistant genotypes were able to maintain plant growth upon virus infection.

At early stages of plant development, screening of root traits can be used as a proxy for mature stages (*Comas et al., 2013*). Greater root length contributed to enhanced grain yield of wheat by permitting more water extraction at grain filling stage (*Manschadi, Christopher & Hammer, 2006*). Barley yellow dwarf virus (BYDV) affected the root elongation in wheat and barley cultivars differing in their response to BYDV, the reduction of total root length was less severe in the resistant cultivars than susceptible cultivars, which is shown in our results as well as previous reports on BYDV infected oat cultivars (*Kolb et al., 1991*). Root length is associated with plant height (*Steele et al., 2006*). Reduction in root length severely restricted water and nutrient absorption process, which may lead to the decreased of shoot growth (*Riedell et al., 2003*).

Long-distance movement of virus particles is known to occur via the phloem, following the stream of sugar transport (*Gilbertson & Lucas, 1996*; *Maule & Palukaitis, 1991*). In sugarcane leaves infected by sugarcane mosaic virus (SCMV), sugar concentration is increased as a result of inhibited phloem transport (*Addy et al., 2017*). *Fiebig, Poehling & Borgemeister (2004)* measured sugar content in phloem sap of BYDV infected and non-infected wheat plants and observed that there was no significant differences between control and infected plants. It is possibly BYDV blocked sugar movement into the phloem and consequently the rate of sugar movement was reduced, thus its concentration within the cells was higher. In this study, leaves infected with BYDV had significantly increased content of reducing sugar in susceptible cultivars. Likewise, *Jensen (1969)* and *Jensen (1968)* showed that BYDV infected plants had greater accumulation of carbohydrate in leaves, and a corresponding reduced chlorophyll content and rate of photosynthesis. The results of our study were similar to those found in a study of sunflower chlorotic mottle virus (SuCMoV) on sunflower in which infected leaves showed higher sugar accumulation and lower shoot biomass (*Arias, Lenardon & Taleisnik, 2003*). *Misra & Jha (1971)* observed an increase in reducing sugar in chilli leaves affected by mosaic virus, as did *Gonçalves et al. (2005)* in sugarcane leaves affected by sugarcane yellow leaf virus (ScYLV), possibly as a result of disruption of normal phloem transport or phloem loading. *Shalitin & Wolf (2000)* found that increased foliar sugar levels in melon plants following infection by cucumber mosaic virus were accompanied by increased respiration, which may lead to biomass reduction.

Protein components has been reported to be involved in plant pathogenic interactions (*Carvalho et al., 2006*; *Tornero et al., 2002*) with BYDV susceptible wheat cultivars showing significant reductions in protein content (*Xu et al., 2016*). In the current experiment, significant reduction in protein content at 6 WAI was also found in resistant cultivars, which is different from the report of *Sahhafi et al. (2012)* that resistant wheat maintained higher protein content under wheat streak mosaic virus infection.

Phenolic compounds are often associated with plant responses to different stresses (*Freeman & Beattie, 2008*), with higher accumulation of phenols in resistant genotypes compared to susceptible ones in other virus/plant interactions (*Siddique et al., 2014*; *Singh et al., 2010*). Deposition of phenolics in plant cell walls might be a possible mechanism of virus resistance by playing key roles in increasing mechanical strength of host cell (*Boudet, Lapierre & Grima-Pettenati, 1995*) and inducing cell wall lignification as lignin precursors (*Lyon et al., 1992*). In the present study, at 3 WAI phenolic content was increased only in virus infected resistant genotypes. In addition, the total phenolic content was significantly increased in all infected resistant genotypes at 6 WAI but not in susceptible ones, suggesting that both increased rate and quantity of phenolics might be components of the defence mechanism of *Bdv2* and *Ryd2* resistance genes.

In a vascular bundle, SE and CC participate in metabolic activities, and are responsible for long distance transport of minerals and assimilates. Viruses have been shown to affect both the structural and functional activities of the SE and CC (*Lalonde, Franceschi & Frommer, 2001*). BYDV is a systemic virus and its replication is almost entirely restricted within the plant phloem tissue (*Irwin & Thresh, 1990*). BYDV particles are found exclusively in vasculature samples (*Gill & Chong, 1975*). Any restriction to phloem tissues will impact

virus dispersal. However, there is also evidence of cell-to-cell movement of luteoviruses between nucleate cells of the phloem tissues (*Mutterer et al., 1999*).

## CONCLUSION

Although the damage to vasculature in BYDV infected plants remains to be quantified, we might speculate that accumulating viral load in the phloem leads to more widespread damage to the vasculature, and inhibition of sugar transport, which in turn inhibits root and biomass growth. The increased sugar content of leaves may also inhibit photosynthesis resulting in a further cycle of growth constraint. We hypothesise that the capacity to respond to virus with inhibitory phenolic compounds may be the basis of *Bdv2* and *Ryd2* resistance, limiting viral load and the cascade of pathological events described above.

## ACKNOWLEDGEMENTS

The authors would like to thanks Dr Fiona Kerslake and Dane Hayes for their help during biochemical analysis and ultrastructure examination respectively.

### Funding
This project was supported by funds from the Grains Research and Development Corporation of Australia. The funders had no role in study design, data collection and analysis, decision to publish, or preparation of the manuscript.

### Grant Disclosures
The following grant information was disclosed by the authors:
Grains Research and Development Corporation of Australia.

### Competing Interests
The authors declare there are no competing interests.

### Author Contributions
- Shormin Choudhury conceived and designed the experiments, performed the experiments, analyzed the data, contributed reagents/materials/analysis tools, prepared figures and/or tables.
- Hongliang Hu performed the experiments, authored or reviewed drafts of the paper.
- Philip Larkin analyzed the data, contributed reagents/materials/analysis tools, authored or reviewed drafts of the paper.
- Holger Meinke and Sergey Shabala authored or reviewed drafts of the paper.
- Ibrahim Ahmed performed the experiments.
- Meixue Zhou conceived and designed the experiments, analyzed the data, contributed reagents/materials/analysis tools, authored or reviewed drafts of the paper, approved the final draft.

## Data Availability

The raw data are provided in the Supplemental Files.

## Supplemental Information

Supplemental information for this article can be found online at http://dx.doi.org/10.7717/peerj.4833#supplemental-information.

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
