# Peer review of "Agronomical, biochemical and histological response of resistant and susceptible wheat and barley under BYDV stress"

_PeerJ, doi:10.7717/peerj.4833_

## Round 0.1 · original submission · Major Revisions

A thorough improvement of the manuscript is necessary before reconsidering it for publication. Especially a better description of the methods (number of plants, replications, etc.) is necessary. Furthermore statistical tests should be employed and described. Please, address all comments of the reviewers in a revised manuscript and describe all changes and additions in an accompanying letter.

Further comments:

Figures 1 and 2: Provide accession names
Figures 3 to 9: Provide statistical tests

·

Basic reporting

No comment

Experimental design

No comment

Validity of the findings

No comment

Additional comments

If there is no special reason, please rearrange figure 1. Change position virus bar and control bar.
All figures exept for figure 1 virus bar locate right side. It is good to maintain consistency for all figures. Please mention meaning of "LISE" at Figure 10-D.

Reviewer 2 ·

Basic reporting

1. This manuscript really needs a narrative, and – to my mind – their experimental work deserves it. The introduction and results are both lists of findings, but ideas are not being developed or connected. In this sense the discussion is better. However, because the paper is not framed around a couple of interesting scientific questions/hypotheses, even the discussion does not feel very coherent or focused. Whilst the findings of the study are not earth shattering and I appreciate the sobriety of their presentation, I’m convinced the authors can find a more thought-provoking and stimulating way of present their results.

2. The paper needs careful proofreading and editing. Articles are often missing, the lit. references are sometimes awkward [… as shown by (X et al., 20NN).] , sentences are incomplete (E.g., second paragraph of the methods section), use of subscripts inconsistent (e.g., OD [subscript] 405).

Experimental design

1. The authors state in the methods how they infected plants for their experiment with aphids, but a lot of details leading up to that point are missing. Where did the aphids come from, what exactly is the source virus, how exactly did the aphids acquire the virus? Very relevant details for understanding their work.
2. How was the cutoff value for the OD405 established? And, given that the ELISA assay has different background levels in the different plants, why would the cutoff be the same?
3. On a related note, are the authors sure all plants were infected? I.e., for Manning, the SE (!) is rather large in Fig. 4, suggesting that not all individual plants have a significant increase in ELISA values. The authors would really need to do a pairwise test to establish that each point is significantly higher than the mock-infected controls. If that’s not true, the uninfected plant(s) should be removed from the analysis.
4. Can the authors give a source for using root length as a proxy for root development? It seems obvious, but since I am not familiar with quantifying root development I wouldn’t know for sure.

Validity of the findings

1. I find it totally unclear how the statistical analyses were performed. What data were analyzed by one-way ANOVA? This is not explained in the methods, and due to the incomplete reporting of statistics in the results (only P-values) it never becomes any clearer. State in the methods exactly what data (virus-infected values normalized by mock-infected controls?) were analyzed with what method, and please justify briefly why this method is suitable (are assumptions being met?). In the results, give full ANOVA tables (preferably in the manuscript proper). It might also be that another statistical test is more suitable here, but this cannot be judged form the information supplied.

Additional comments

Choudhury and colleagues present a study the effects of BYDV-PAS infection on two cultivars of wheat and barley, focusing on virus effects on host growth, photosynthesis, the presence of different biomolecule and leaf structure. They find the expected differences between the two cultivars – with different susceptibilities – taken for each species. I don’t see any major problems with the experimental study, although I find the description of the statistical methods and results very unclear. Finally, right now the manuscript is more a technical report than a scientific paper: there are no hypotheses, it is not clear what exactly this whole exercise teaches us, and there is no real narrative. I given some further detailed comments I hope the authors will find helpful.

Reviewer 3 ·

Basic reporting

This paper presents data on the effect of BYDV-PAV infection on resistance or susceptible wheat and barley cultivars. There is no justification of the election of the analysed parameters (biomass, root length, chlorophyll context etc), and it is unclear how or why the effect of infection on these parameters may help understand resistance mechanisms. Resistance mechanisms are better analysed with the tools and approaches of molecular genetics. In addition, the methodology is not properly described nad there are some conceptual errors.
Specific comments:
1. The Introduction does not lead anywhere nor indicates why the study is made as it is. There is no connection between the paragraphs relating to vector transmission (not analysed) phenolic content or sugar content.
2. In addition most references are quite old and should be updated. I suggest visiting Volume 17 of Current Opinion in Virology (2016) dedicated t plant virus pathogenesis.
3. The assertion lines 49-50 is not correct, as it only applies to persistently transmitted viruses.
4. There is no indication of the source, genotype or isolate of the virus used. Same for the aphids.
5. The experimental design is not explained: what is considered as a replica? Apparently not a single plant, what then, a pot? With how many plants? Etc
6. Data were analysed by ANOVA. Were data distributed normally and homoscedastically for this analysis to be valid? How were comparisons between treatments done? No information on sigificnace of differences is given in figures or text.
7. Figures are miss-numbered. Fig. 2 is unnecessary.
8. Tolerance and resistance are different defences in plants, But the cultivars here used are sometimes rated as resistant and sometimes as tolerant. What are they? According to Fig. 4, it seems they are resistant, as virus accumulation is decreased.
9. On which data are assertions on ultrastructure done? Fig. 10 does not show electron micrographs.
10. Discussion is mostly a repetition of the results.

Experimental design

There is no information on the experimental design.

Validity of the findings

Cannot know, as statical analyses are not presented and the experimental design is not described.

Additional comments

This paper presents data on the effect of BYDV-PAV infection on resistance or susceptible wheat and barley cultivars. There is no justification of the election of the analysed parameters (biomass, root length, chlorophyll context etc), and it is unclear how or why the effect of infection on these parameters may help understand resistance mechanisms. Resistance mechanisms are better analysed with the tools and approaches of molecular genetics. In addition, the methodology is not properly described nad there are some conceptual errors.

Reviewer 4 ·

Basic reporting

See comments below

Experimental design

See comments below

Validity of the findings

See comments below

Additional comments

Comments to the Manuscript PeerJ#22841
* For better understanding please give some information to the terms ‚virus resistance‘ and ‚virus tolerance‘ (e.g. in the introduction).
* Please use these terms clearly (precisely) in the whole manuscript, compare e.g.

L 87 …cultivars with known BYDV resistance…

L 209 …two tolerant cultivars…
* Please give some information about the known effects of the Ryd2 and Bdv2 genes.
* Material and methods
Give some more information to the method of ELISA, e.g. how many positive and negative controls did you use? How did you estimate the detection threshold?
* Please discuss your results more in detail also in relation to prior results (e.g. Jensen 1968, 1969, Orlob and Arny, 1961 …) to bring out the new findings of your studies.
* Please improve the legends of the figures and check Fig. 1 – what is presented?
* The style of professional English has to be improved.
* Please find more comments in the attached file (PeerJ#22841.pdf) of the manuscript.

Annotated reviews are not available for download in order to protect the identity of reviewers who chose to remain anonymous.

---

## Round 0.2 · Minor Revisions

The revised manuscript has improved, but there are still some concerns and questions of a reviewer. Address those in a further Revision of the manuscript and clearly indicate in the accompanying letter in which lines of the manuscript the respective Revisions were performed.

Figures 3 to 9: indicate significance threshold of asterisks in the legends.
Why were there different n for the different experiments?

Reviewer 2 ·

Basic reporting

Given there are still typos in the ms, a final round of proofreading would probably be a good thing.

Experimental design

I previously commented on the experimental design:

“3. On a related note, are the authors sure all plants were infected? I.e., for Manning, the SE (!) is rather large in Fig. 4, suggesting that not all individual plants have a significant increase in ELISA values. The authors would really need to do a pairwise test to establish that each point is significantly higher than the mock-infected controls. If that’s not true, the uninfected plant(s) should be removed from the analysis.”

Compared to Fig. 4 in the original submission, virus bar for “Manning” now has a higher mean and smaller standard error. I can’t follow from the authors’ very brief comment in the rebuttal or the revised manuscript why this is. Were some of the plants indeed not infected and therefore removed? And where these plants then also removed from the other analyses? I think this information should be provided in the manuscript.

Validity of the findings

Under validity of the findings I previously asked about the use of ANOVA. The authors acknowledge this comment, and state they clarified this point in the methods. However, homoscedasticity is not discussed anywhere and the new Table 1 with only F-values is not very helpful. So I’m afraid I have to be very boring and repeat myself: address whether the data meet assumptions for ANOVA and please present full ANOVA results (Tables in Supp. Materials and F test results in paper proper?).

Additional comments

The authors have revised their manuscript and taken on board some of the criticisms. Importantly, the narrative has been improved. However, some aspects are still opaque to me.

Reviewer 4 ·

Basic reporting

'no comment'

Experimental design

'no comment'

Validity of the findings

The results of ANOVA (table 1) should be discussed in connection with the different analysed parameters, not in a separate paragraph (line 293-296).

Table 1 as supplement?

Additional comments

Proofreading of the whole manuscript is necessary (see also attached file).

Annotated reviews are not available for download in order to protect the identity of reviewers who chose to remain anonymous.

---

## Round 0.3 · accepted · Accept

The revised version of the manuscript can be accepted for publication.

#